# Anticancer Effects of 6-Gingerol through Downregulating Iron Transport and PD-L1 Expression in Non-Small Cell Lung Cancer Cells

**DOI:** 10.3390/cells12222628

**Published:** 2023-11-15

**Authors:** Dong Young Kang, Sanghyeon Park, Kyoung Seob Song, Se Won Bae, Jeong-Sang Lee, Kyoung-Jin Jang, Yeong-Min Park

**Affiliations:** 1Department of Immunology, School of Medicine, Konkuk University, Chungju 27478, Republic of Korea; 2Department of Medical Science, Kosin University College of Medicine, Busan 49267, Republic of Korea; 3Department of Chemistry and Cosmetics, Jeju National University, Jeju 63243, Republic of Korea; 4Department of Functional Foods and Biotechnology, College of Medical Sciences, Jeonju University, Jeonju 55069, Republic of Korea; 5Department of Bioscience and Biotechnology, Konkuk University, Seoul 05029, Republic of Korea; 6Department of Integrative Biological Sciences and Industry, Sejong University, Seoul 05006, Republic of Korea

**Keywords:** NSCLC, 6-gingerol, iron metabolism, oxidative stress, EGFR/JAK2/STAT5b signaling, PD-L1, miR-34a/miR-200c

## Abstract

Iron homeostasis is considered a key factor in human metabolism, and abrogation in the system could create adverse effects, including cancer. Moreover, 6-gingerol is a widely used bioactive phenolic compound with anticancer activity, and studies on its exact mechanisms on non-small cell lung cancer (NSCLC) cells are still undergoing. This study aimed to find the mechanism of cell death induction by 6-gingerol in NSCLC cells. Western blotting, real-time polymerase chain reaction, and flow cytometry were used for molecular signaling studies, and invasion and tumorsphere formation assay were also used with comet assay for cellular processes. Our results show that 6-gingerol inhibited cancer cell proliferation and induced DNA damage response, cell cycle arrest, and apoptosis in NSCLC cells, and cell death induction was found to be the mitochondrial-dependent intrinsic apoptosis pathway. The role of iron homeostasis in the cell death induction of 6-gingerol was also investigated, and iron metabolism played a vital role in the anticancer ability of 6-gingerol by downregulating EGFR/JAK2/STAT5b signaling or upregulating p53 and downregulating PD-L1 expression. Also, 6-gingerol induced miR-34a and miR-200c expression, which may indicate regulation of PD-L1 expression by 6-gingerol. These results suggest that 6-gingerol could be a candidate drug against NSCLC cells and that 6-gingerol could play a vital role in cancer immunotherapy.

## 1. Introduction

Several natural substances have anticancer potential against various cancers with minimal side effects to counter the side effects of chemotherapy medications [1,2,3]. Natural-compound cancer treatments are a good alternative to chemotherapeutics because they can provide multitargeted treatment with fewer adverse effects [4,5]. Ginger is a widely used Asian spice, and 6-gingerol is a major pharmacological constituent [6], with antioxidant [7], antiplatelet [8], anti-inflammatory [9], antiproliferative [10,11], and anticancer [12] properties. 6-Gingerol has anticancer properties against various malignancies, including pancreatic, gastric, colorectal, renal, and breast cancers [13,14,15]. A recent study on embryonic cancer stem cells also indicated the regulation of iron metabolism by 6-gingerol. Hence, understanding the exact role of 6-gingerol in iron homeostasis in non-small cell lung cancer (NSCLC) will be interesting. NSCLC is known to comprise nearly 80% of all lung cancer cases. *ERBB2*, *EGFR*, and *KRAS* are generally the top three mutation genes occurring in NSCLC. These mutations commonly lead to uncontrollable cell growth, metastasis, and drug resistance. Kirsten rat sarcoma viral oncogene homolog (KRAS) mutation is especially prevalent in NSCLC cell lines, including A549 and H460 cells, and occurs in up to 30% of all cases [16]. Mutations in KRAS are known to weaken considerably the therapeutic effects of drugs in NSCLC patients [17,18]. KRAS has been commonly used in a challenging therapeutic target over the past few decades [19].

Iron is an essential cofactor in enzymes that control cell development and death, serves as an important part of cellular metabolism [20], and plays a role in DNA synthesis, damage response, and mitochondrial function. Iron is a necessary component of the human body and plays a critical function in oxygen delivery [21]. Furthermore, reactive oxygen species (ROS) are produced during biomolecule oxidation, making iron hazardous [22,23]. Iron homeostasis must use iron channels to balance the amount of iron to prevent excessive ROS. Iron may have a role in cancer by preserving genomic integrity and regulating epigenetic changes and plays a role in the tumor microenvironment and metastasis [24]. The possibility that iron homeostasis has a dual role in cancer cell death and tumor development is contingent on how it acts in cells. Macrophages are also involved in iron metabolism. M1 macrophages use heme oxygenase-1 and ferroportin (FPN) to accumulate iron [25], whereas M2 macrophages use ferritin to diminish iron levels by exporting and inhibiting proinflammatory cytokines [26]. Also, transferrin receptor 1 (TFR1) allows transferrin–iron complexes (Fe^3+^) to enter the cytoplasm during iron transport. Iron is then transformed into ferrous ions (Fe^2+^) by enzymes, such as six-transmembrane epithelial antigen of prostate 3 (STEAP3) and divalent metal transporter 1 (DMT1), before entering cellular metabolism and heme production [27], emphasizing the importance of iron metabolism in the body because it can lead to inflammation and tumorigenesis if it is disrupted.

Epidermal growth factor receptor (EGFR) is a driver of tumorigenesis; after phosphorylation, EGFR promotes tumor development by activating proto-oncogenes [28]. EGFR levels are elevated in various human malignancies, allowing signal transduction in the cell cytoplasm and promoting carcinogenesis. EGFR may have a role in altering cellular phenotypes that promote tumor cell survival and proliferation [29]. The ligand connects to the extracellular side of a transmembrane receptor and starts a cascade of phosphorylation reactions in the JAK-STAT pathway. Phosphorylation of JAK kinases, the receptor cytoplasmic tail, and STAT transcription factors that phosphorylate STATs promote nucleoplasm localization, DNA binding, and gene regulation during tumorigenesis [30]. Thus, several methods have been developed to produce treatment by inhibiting STAT5 phosphorylation; the therapeutic benefits of reducing its activity in cancer have been proven in vitro and in vivo [31].

The PD-1/PD-L1 complex plays a key role in tumor proliferation by disrupting growth regulation, leading to a deficiency in programmed cell death or apoptosis [32]. Furthermore, a high amount of PD-L1 has been observed on the surface of many cancer cells, including NSCLC, suggesting that its expression aids cancer cells in evading immune responses [2]. JAK/STAT pathways are important to producing PD-L1 in tumor cells involved in various cancers [33]. Moreover, antitumor immune responses are suppressed by the PD-1/PD-L1 inhibitory checkpoint. In many studies, anti-PD-1/PD-L1 therapy that suppresses its signaling is effective and promising in metastatic NSCLC [34]. The suppression of EGFR signaling can activate the immune system by decreasing PD-L1 production via activation of p53 signaling, which plays a vital role in apoptosis induction [35].

MicroRNAs (miRNAs) can impede mRNA translation and increase mRNA breakdown, resulting in post-transcriptional gene expression changes. By interacting with PD-L1, these miRNAs also have a role in the tumor immune system [36]. As a result, miR-34a and miR-200c are linked to PD-L1 production in response to p53-dependent antitumor actions [37,38]. Although multiple studies have shown that miRNA directly influences PD-L1 expression, little investigation exists into the impacts generated indirectly by p53 through medications, including natural compounds. This study investigated the role of iron metabolism in the anticancer activity of 6-gingerol in NSCLC cells.

## 2. Materials and Methods

### 2.1. Reagents and Antibodies

RPMI-1640 medium was purchased from Gibco (Thermo Fisher Scientific, Inc., Waltham, MA, USA). 6-Gingerol (catalog no. 23513-14-6) was purchased from TCI (Tokyo Chemical Industry Co., Tokyo, Japan). miRNA inhibitor (#AM17000) was purchased from Thermo Fisher Scientific, Inc. Fetal bovine serum (FBS; 12003C) and iron (II) sulfate heptahydrate (F8633) were purchased from Sigma-Aldrich (Merck KGaA, St. Louis, MO, USA). Antibodies specific for cyclin E (sc-481), p21 (sc-756), Bcl-2 (sc-7382), and CDK4 (sc-260) were obtained from Santa Cruz Biotechnology, Inc. BCL-xL (#2764), p27 (#3686), BAX (#2772), p53 (#9282), cytochrome c (#11940), pBRCA1 (#9009), pCHK1 (#2348), pATM (#5883), pATR (#2853), pJAK2 (#3776), pCHK2 (#2197), C-Casp9 (#9505), pSTAT5 (#9351), JAK2 (#3230), COX IV (#4850), and GAPDH (#2118) antibodies were purchased from Cell Signaling Technology, Inc., Danvers, MA, USA. STEAP3 (ab151566), TFR1 (ab84036), cyclin D1 (ab6152), and DMT1 (ab55735) antibodies were purchased from Abcam, Cambridge, UK. FPN1 (NBP1-21502) and iNOS (NB300-650) were obtained from Novus Biologicals. The antibody specific for PD-L1 (R30949) was obtained from NSJ Bioreagents.

### 2.2. Cell Culture

A549 (KCLB No. 10185) and H460 (KCLB No. 30177) cell lines were purchased from the Korean Cell Line Bank (Jongno-gu, Seoul, Korea). Cells were cultured and maintained in RPMI-1640 medium plus 10% FBS and 1% penicillin at 37 °C in 5% CO_2_. Treated cells were incubated at 37 °C for 48 h.

### 2.3. MTT Assay

The MTT assay was used to measure cell viability. The formazan product absorbance was measured with a microplate at a 560 nm wavelength.

### 2.4. Real-Time qPCR

Using the TRIzol method, total RNAs were isolated. The qPCR was performed using LightCycler 480II (Roche). The primer sequences are described in Appendix A. This assay was carried out as described previously [2].

### 2.5. Immunoblotting Assay

Proteins were isolated from cells using RIPA lysis buffer (20–188; EMD Millipore, Merk Millipore, Burlington, MA, USA). This assay was carried out as described previously [2].

### 2.6. Mitochondrial Membrane Potential (MMP) and ROS Analysis

Cultured cells were harvested, and this assay was carried out as described previously [1].

### 2.7. Cell Cycle Analysis

The DNA content of 6-gingerol-treated and nontreated cells was determined using the BD Cycletest Plus DNA Reagent Kit (BD Biosciences, San Diego, CA, USA: #340242) following the manufacturer’s protocol, essentially as described previously [1].

### 2.8. Comet Assay

A comet assay kit (Abcam; #ab238544) was used to measure cellular DNA damage, essentially as described previously [1].

### 2.9. Apoptosis Analysis

The Annexin V Conjugates for Apoptosis Detection kit (Sigma-Aldrich, Merck KGaA; #A13199) was used to measure apoptosis in A549 or H460 cells according to the manufacturer’s instructions.

### 2.10. Isolation of Mitochondria/Cytosol Fractions

Mitochondria/cytosol fractions from 6-gingerol-treated and nontreated A549 or H460 cells were extracted using a mitochondria/cytosol fractionation kit (Cambridge, MA, USA, Abcam; ab65320) as described previously [30].

### 2.11. ATP Determination Assay

An ATP determination kit (Molecular Probes, Eugene, OR, USA) was used to measure ATP. Briefly, A549 or H460 cells were treated with 6-gingerol, and an equal number of cells was collected for ATP determination assay, essentially as described previously [30].

### 2.12. Iron Estimation Assay

Iron estimation was performed using an iron assay kit (MAK025) purchased from Sigma-Aldrich (Merck KGaA). This assay was carried out as described previously [30].

### 2.13. FACS Analysis for Ferrous Iron

After cultured cells were washed with culturing medium, a 2 mL staining solution containing FerroFarRed (5 µM; GC903-01; Goryo Chemical, Sapporo, Japan) was added. Further incubation was performed in a CO_2_ incubator at 37 °C for 30–40 min. Cells were washed with 1 mL PBS after staining and used for FACS analysis.

### 2.14. Statistical Analyses

All experiments were performed in triplicate. Results were expressed as the mean ± standard error of the mean. Statistical analyses were conducted using one-way analysis of variance (ANOVA) or Student’s *t*-test. Additionally, one-way ANOVA was performed using Tukey’s post hoc test. Analyses were performed using SAS 9.3 software (SAS Institute, Inc., Cary, NC, USA). *p* < 0.05 (*) was considered statistically significant.

## 3. Results

### 3.1. 6-Gingerol Induced ROS Generation and DNA Damage Repair in NSCLC Cells

The molecular mechanism of the anti-cancer activities of 6-gingerol was investigated. First, a possible ROS generation was assumed to have been key to the inhibitory activity against cancer cell proliferation by 6-gingerol. To analyze this, iNOS expression with or without 6-gingerol in protein levels was first determined (Figure 1A). The results suggested increased iNOS expression with increasing 6-gingerol concentrations in NSCLC cells. This expression pattern in mRNA levels was performed by RT-PCR, and the results indicate a significant upregulation in iNOS expression upon 6-gingerol treatment (Figure 1B). These results offer a hint of a possible ROS generation, and the hypothesis was confirmed by checking the cellular ROS (Figure 1C) and mitochondrial ROS (Figure 1D) levels in A549 and H460 cells after treatment with 200 µM 6-gingerol. These results support the hypothesis and the antiproliferative capacity of 6-gingerol in two NSCLC cells.

The ability of 6-gingerol to induce DNA damage response (DDR) against NSCLC cells was also investigated. The proteins responsible for DDR with or without 6-gingerol treatment were analyzed to determine the capability of 6-gingerol to induce DDR. These results suggest upregulated pATM, pATR, pCHK1, pCHK2, and pBRCA1 expression upon 100 and 200 µM 6-gingerol treatment in A549 and H460 cells (Figure 1E). DNA double-stranded breaks in these NSCLC cells and their induction after 6-gingerol treatment were also investigated. Moreover, 6-gingerol enforced double-stranded breaks in these NSCLC cells with the comet assay (Figure 1F). 6-Gingerol treatment enhanced the comet length and the comet-positive cells (Figure 1G). These results indicate the primary mechanism of the antiproliferative activity of 6-gingerol against NSCLC cells.

### 3.2. 6-Gingerol Induced p53 Expression and Arrest of Cell Cycle in NSCLC Cells

6-Gingerol inhibited proliferation of cancer cell via ROS generation and DDR induction. These results led to an assumption of cell cycle arrest induction by 6-gingerol in NSCLC cells, as prolonged DNA damage leads to uncontrolled cell cycle divisions. Analysis of the cell cycle was performed to analyze this hypothesis, and the results show an arrest in the G0/G1 phase upon 6-gingerol treatment (Figure 2A). Molecular analysis of proteins responsible for the cell cycle upon 6-gingerol treatment was also conducted. The results suggest upregulated p53 expression in 6-gingerol NSCLC cells (Figure 2B), which is a key protein for deciding cell fate. Increased cell cycle checkpoints such as p21 and p27 were also detected upon 6-gingerol treatment with decreased protein expression of cell cycle promoters CDK4, cyclin D1, and cyclin E. Induction of cell cycle arrest by 6-gingerol in mRNA levels was confirmed, and the results show similar expression patterns to protein levels, which provides strong support to the hypothesis of the role of 6-gingerol in inducing cell cycle arrest (Figure 2C).

### 3.3. 6-Gingerol Induced Mitochondrial Apoptosis in NSCLC Cells

Cell proliferation analysis with or without 6-gingerol treatment in A549 and H460 cell lines was performed to identify the anticancer abilities of 6-gingerol against NSCLC cells. Cell viability assay with increasing concentrations of 6-gingerol (50–400 µM) showed significant inhibition in cell proliferation in proportion to concentration (Figure 3A). 6-Gingerol (200 µM) induced ~50% cell death against both cell lines, and this concentration was chosen as the IC_50_ dosage in this study. However, 6-Gingerol (200 µM) showed lower cytotoxicity in two non-cancerous HEK 293T and C2C12 cells (Appendix A). Totals of 100 and 200 µM 6-gingerol were used in most studies to analyze the increasing concentration effect. Cell death and morphology were also analyzed using microscopy. Brightfield microscopy of cells displayed a considerable decrease in cell number, confirmed via nuclei count using a DAPI staining assay (Figure 3B). 6-Gingerol was shown to have antiproliferative ability against NSCLC cells.

The antiproliferative activity of 6-gingerol was observed by generating ROS and inducing DDR in NSCLC cells, suggesting a possible apoptosis induction in these cancer cells by 6-gingerol. Apoptosis was first investigated using flow cytometry to determine cell death induction by 200 µM 6-gingerol, and the results suggest a slight apoptosis induction by 6-gingerol in A549 cells and more prominent apoptosis in H460 cells (Figure 3C). The molecular patterns of proteins responsible for apoptosis with or without 6-gingerol were investigated against A549 and H460 cells because there was a slight apoptosis induction with 6-gingerol (Figure 3D). The results show downregulated BCL-xL and BCL-2 expression and upregulated BAX expression with increasing concentrations of 6-gingerol, indicating mitochondrial membrane integrity loss with 6-gingerol treatment. The expression of these genes in mRNA levels with 6-gingerol was also confirmed, and the results show a similar pattern to that of protein levels (Figure 3E). A decrease in BCL-2 and an increase in BAX expression indicated pore formation in the mitochondrial membrane to release cytochrome c from mitochondria to cytosol.

MMP was determined to check the mitochondrial membrane integrity upon 6-gingerol treatment, and the results suggest a loss of MMP in 6-gingerol-treated cells compared to nontreated control cells (Figure 3F). Loss of MMP also led to a loss of ATP in cancer cells, and the results for ATP formation in A549 and H460 cells also show decreased ATP concentration upon 6-gingerol treatment (Figure 3G). These results suggest the ability of 6-gingerol to induce mitochondrial apoptosis and that cytochrome c release plays a vital role in this process. Cytochrome c expression in A549 and H460 cells was upregulated after treatment with increasing concentrations of 6-gingerol, suggesting cytochrome c release through mitochondrial pore formation due to BAX/BCL-2 regulation by 6-gingerol (Figure 3H). Cytosol and mitochondrial fractionation and cytochrome c expression analysis with or without 6-gingerol were performed to confirm cytochrome c release from mitochondria to the cytosol (Figure 3I). This result lends strong support to the role of 6-gingerol in inducing the intrinsic apoptosis pathway against NSCLC cells.

### 3.4. 6-Gingerol Suppressed Cancer Stemness and Tumor Invasion in NSCLC Cells

6-Gingerol can inhibit the proliferation of cancer cells by generating ROS and activating DDR, cell cycle arrest, or mitochondrial apoptosis. These abilities of 6-gingerol led to the investigation into cancer stemness and tumor metastasis. Tumorsphere formation in A549 and H460 cells with or without 6-gingerol treatment was investigated to analyze the cancer stemness (Figure 4A). The results show large tumorsphere formation in nontreated control cells after 7 days, whereas 200 μM 6-gingerol drastically suppressed tumorsphere formation. These results suggest the ability of 6-gingerol to act against cancer stem cells within cancer cells. The capability of 6-gingerol to act against tumor metastasis was investigated by conducting a cell invasion assay in NSCLC cells (Figure 4B). These results show fewer invaded cells in 6-gingerol-treated cells compared to nontreated control cells. As tumor invasion is a key process in tumor metastasis, inhibition of tumor invasion by 6-gingerol shows a hint of its antimetastatic ability against two NSCLC cells.

### 3.5. 6-Gingerol Regulated Iron Homeostasis and Acted as a Key Mechanism in Apoptosis Induction

The anticancer activity of 6-gingerol regulates various cancer hallmarks in NSCLC cells. However, the mechanism of the ability of 6-gingerol in NSCLC is still unknown. Iron was assumed to potentially play a key role in apoptosis induction by 6-gingerol. First, the expression of proteins responsible for iron transport with or without 6-gingerol treatment was investigated in NSCLC cells. The result indicate a concentration-dependent inhibition in STEAP3, DMT1, FPN1, and TFR1 expression upon 6-gingerol treatment, suggesting the capacity of 6-gingerol to suppress iron transport in cancer cells (Figure 5A). As these results indicate the role of iron homeostasis inhibition by 6-gingerol, an iron release assay was conducted in NSCLC cells with or without 6-gingerol treatment, and the results show a decreased number of total iron in cells and an increased amount of total iron in the spent medium (Figure 5B). These results suggest the release of iron from cells to the medium by 6-gingerol, supporting previous results of the inhibition of proteins responsible for iron transport. For further confirmation, Fe^2+^ levels in NSCLC cells upon 6-gingerol treatment were determined using flow cytometry (Figure 5C). The results show decreased expression of ferrous ions by 6-gingerol treatment, clearly suggesting the inhibition of the conversion from ferric ions (Fe^3+^) to ferrous ions. These results suggest the role of iron homeostasis in the anticancer activity of 6-gingerol in NSCLC cells.

### 3.6. 6-Gingerol Inhibited the EGFR/JAK2/STAT5b Pathway and PD-L1 Signaling in NSCLC Cells

6-Gingerol has antiproliferative activity against NSCLC cells, and the mechanism of cell death induction by 6-gingerol is mediated by iron homeostasis inhibition. The regulation of various molecular targets by 6-gingerol on its anticancer activity was also observed. Various molecular pathways reported that the EGFR/JAK2/STAT5b pathway was included in antiproliferative activity of 6-gingerol (Figure 6A). The results show the suppression of pEGFR, pJAK2, and pSTAT5 expression without affecting the expression of their respective total forms. PD-L1 expression was also inhibited, hinting at the possible mechanism of the EGFR/JAK2/STAT5b pathway that leads to PD-L1 expression, which was successfully inhibited by 6-gingerol treatment. As STAT5b could act as a transcription factor for PD-L1, STAT5b binding to PD-L1 promoter may be interesting in the case of antiproliferative activity of 6-gingerol. A chromatin immunoprecipitation (ChIP) assay was conducted to check the binding of STAT5b to the promoter of PD-L1. The results showed a significant suppression in the STAT5b/PD-L1 complex with 6-gingerol in A549 and H460 cells (Figure 6B). Also, mRNA levels of PD-L1 were downregulated by 6-gingerol in both NSCLC cells (Figure 6C). These results suggest the inhibition of the EGFR/JAK2/STAT5b pathway and PD-L1 expression by 6-gingerol in NSCLC cells. The role of iron metabolism in this signaling pathway was investigated by analyzing the expression of these signaling molecules in the presence of iron sulfate (FeSO_4_) and 6-gingerol (Figure 6D). Increased pEGFR and PD-L1 expression and decreased p53 expression were noted upon FeSO_4_ treatment, suggesting the role of iron metabolism in the expression of this signaling pathway. 6-Gingerol reversed the effect of FeSO_4_, suggesting the involvement of iron homeostasis in the antiproliferative activity of 6-gingerol against NSCLC cells.

### 3.7. 6-Gingerol Elevated the Expression of miR-34a/miR-200c Signaling in NSCLC Cells

miRNAs are known to play an important role in PD-L1 expression or its inhibition by 6-gingerol because 6-gingerol inhibited PD-L1 expression in A549 and H460 cells. Among various miRNAs, miR-200c and miR-34a play a pivotal function in PD-L1 expression, and 6-gingerol enhanced miR-200c and miR-34a expression (Figure 7A), which correlates with the inhibition of PD-L1 expression. miR-34a and miR-200c expression was inhibited with miRNA inhibitor treatment, confirming the ability of 6-gingerol to upregulate the expression of these miRNAs, and 6-gingerol addition significantly upregulated the expression of these miRNAs (Figure 7B). Furthermore, FeSO_4_ addition decreased miR-34a and miR-200c expression, and 6-gingerol treatment significantly elevated the expression of these miRNAs (Figure 7C). These results may imply that 6-gingerol inhibits PD-L1 expression through the upregulation of miR-34a and miR-200c expression, indicating the antiproliferative activity of 6-gingerol in NSCLC cells.

## 4. Discussion

Studies on the anticancer activity of natural compounds are always a trending field because these compounds lack serious side effects and their ability for multitargeted treatment could help researchers invest more in their mode of mechanism against various cancer types [1,2]. The usage of natural compounds also facilitates the long-term exposure of selected drugs, which helps to target cancer cells and cancer stem cells. This also discourages the possibility of cancer recurrence and thereby suppresses other key cancer hallmarks. Studies on 6-gingerol have already proven its anticancer activity, but its mechanism of action is still under investigation. The anticancer activity of 6-gingerol in A549 and H460 NSCLC cells was confirmed by its ability to inhibit the cell proliferation of these cancer cells. The potential of 6-gingerol to act against cancer stemness and tumor metastasis was also investigated. Inhibition of tumorsphere formation by 6-gingerol suggests its capacity to target cancer stem cells, and suppression of tumor invasion by 6-gingerol indicates its antimetastatic activity against aggressive NSCLC cells. These results could help consider 6-gingerol as a candidate against NSCLC cells.

A candidate drug for anticancer activity should have the ability to mediate various cancer hallmarks, especially apoptosis induction [3]. The drug could be considered a potential candidate to proceed with further anticancer studies if a natural compound exhibits DDR, cell cycle arrest, and apoptosis against cancer cells [13]. 6-Gingerol treatment induced iNOS expression and thereby ROS production, with the induction of iNOS expression directly correlating with ROS generation [39]. A similar pattern was discovered with 6-gingerol treatment in NSCLC cells, which played a vital role in the antiproliferative activity. Initiating DNA double-stranded breaks in NSCLC cells also played a central role in the antiproliferative capability of 6-gingerol, and upregulation of the key DNA damage-sensing kinases such as ATR and ATM by activating CHK signaling also suggests the ability of 6-gingerol to induce DDR against these NSCLC cells [40]. Prolonged DNA damage is believed to enhance uncontrolled cell division and cell proliferation of cancer cells [41]. Hence, DDR also directs an arrest in cell cycle progression. Based on this assumption, 6-gingerol also exhibited the ability to induce cell cycle arrest in A549 and H460 cells by regulating cell cycle checkpoints cyclin D1, cyclin E, CDK4, p21, and p27. Tumor suppressor gene p53 is central to these mechanisms, including the regulation of DNA damage, cell cycle progression, and control over apoptosis induction [42]. Elevated p53 expression upon 6-gingerol treatment also supported its claim for anticancer activity.

Mitochondria are the deciding factor for cell death in cancer cells because they produce ATP as a chemical source of energy for their well-known powerhouse activity [43]. Mitochondrial membrane disruption could facilitate the loss of ATP formation and thereby induce cell death or apoptosis [44]. 6-Gingerol indicated a loss in integrity of the mitochondrial membrane and diminished ATP formation. This process pointed toward mitochondria-dependent apoptosis induction upon 6-gingerol treatment. BAX and BCL-2 regulate mitochondrial membrane integrity, and an alteration in the BAX/BCL-2 ratio defines the cell fate, as increased BAX expression and decreased BCL-2 expression lead to pore formation in the mitochondrial membrane, encouraging cytochrome c release from the mitochondria to the cytosol to regulate caspase activity to proceed with apoptosis [45,46]. Results of 6-gingerol in A549 and H460 cells show increased BAX expression and decreased BCL-2 expression upon release of cytochrome c from the mitochondria to cytosol and the induction of C-Casp9 expression, which proceeds to the intrinsic apoptosis pathway.

In cancer cells, iron homeostasis is strongly attached to ROS generation and DDR induction because iron is considered an essential nutrient with complex activity in tumor biology [47,48]. Iron transport regulation could be the key to anticancer activity because iron metabolism mediates tumor progression. TFR1 and FPN1 mediate homeostasis through iron transport to maintain iron homeostasis, where TFR1 takes up Fe^3+^ for metabolism and FPN1 exports this iron outside [49]. DMT1, also a membrane transporter of iron, helps with intestinal iron absorption via Fe^3+^ reduction before Fe^2+^ translocation [50]. STEAP3 also plays a vital role in iron transport by reducing released Fe^3+^ to Fe^2+^ inside the endosome [51]. The results of 6-gingerol in A549 and H460 cells show a decrease in iron inflow to cells and inhibited Fe^2+^ formation. Downregulation of intracellular Fe^2+^ levels by 6-gingerol is considered the intrinsic apoptosis in A549 and H460 cells. Downregulation of STEAP3, DMT1, FPN1, and TFR1 expression upon 6-gingerol treatment also indicates the role of iron metabolism in the anticancer ability of 6-gingerol.

PD-L1 is a transmembrane protein that plays a pivotal role in immune escape mechanisms by binding to the PD-1 ligand in immune cells. Also, it participates in cancer cell proliferation; hence, PD-L1 expression is considered an interesting target in anticancer activity studies [52]. EGFR/JAK2/STAT5b is also included in PD-L1-dependent cancer cell proliferation, and targeting this signaling pathway in NSCLC could be interesting due to EGFR overexpression, which is highly associated with many NSCLC cells, including A549 and H460 [53]. These results also indicate an inhibition in the expression of this pathway activation along with PD-L1 expression by 6-gingerol, and iron metabolism and p53 expression are also anticancer mechanisms of 6-gingerol.

The many miRNAs are known to mediate PD-L1 expression in direct and indirect ways, with negative and positive regulatory effects [54]. These small coding RNAs could mediate the post-transcriptional modification by regulating mRNA translation and degradation [55]. miR-34a and miR-200c negatively regulate PD-L1 expression, and p53 acts as a mediator in PD-L1 expression, along with miR-34a [56]. miR-200 is a known mediator of cancer cell epithelial-to-mesenchymal transition and is associated with increased tumor cell expression of PD-L1 [57]. miR-34a and miR-200c expression increased upon 6-gingerol treatment even after inhibiting their expression with an miRNA inhibitor. The role of iron metabolism in the inhibition of PD-L1 expression was also proven by regulating the expression of these miRNAs. The results of FeSO_4_ supplementation indicate decreased miR-200c and miR-34a expression, which was considerably increased with 6-gingerol treatment, providing evidence for the role of iron metabolism in the anticancer activity of 6-gingerol.

In this study, we used A549 and H460 cells harboring KRAS mutation. Although there is a lack of studies of genetic backgrounds such as KARS mutation by 6-gingerol, our results indicate that 6-gingerol has anticancer effects, which include the induction of intrinsic apoptosis and the inhibition of cell cycle arrest, invasion, and tumorsphere formation via downregulating intracellular iron levels and PD-L1 expression in A549 and H460 NSCLC cells.

## 5. Conclusions

The ability of 6-gingerol to enhance mitochondrial apoptosis in two NSCLC cells was determined via ROS generation, DDR, and cell cycle arrest induction. Iron metabolism plays a vital role in the anticancer activity of 6-gingerol by regulating the EGFR/JAK2/STAT5b pathway and mediating PD-L1 expression. Iron homeostasis also regulates miR-34a/miR-200c expression to inhibit PD-L1 expression upon 6-gingerol treatment, suggesting the role of iron metabolism in the anticancer ability of 6-gingerol against NSCLC cells (Figure 8).

## Figures and Tables

**Figure 1 cells-12-02628-f001:**
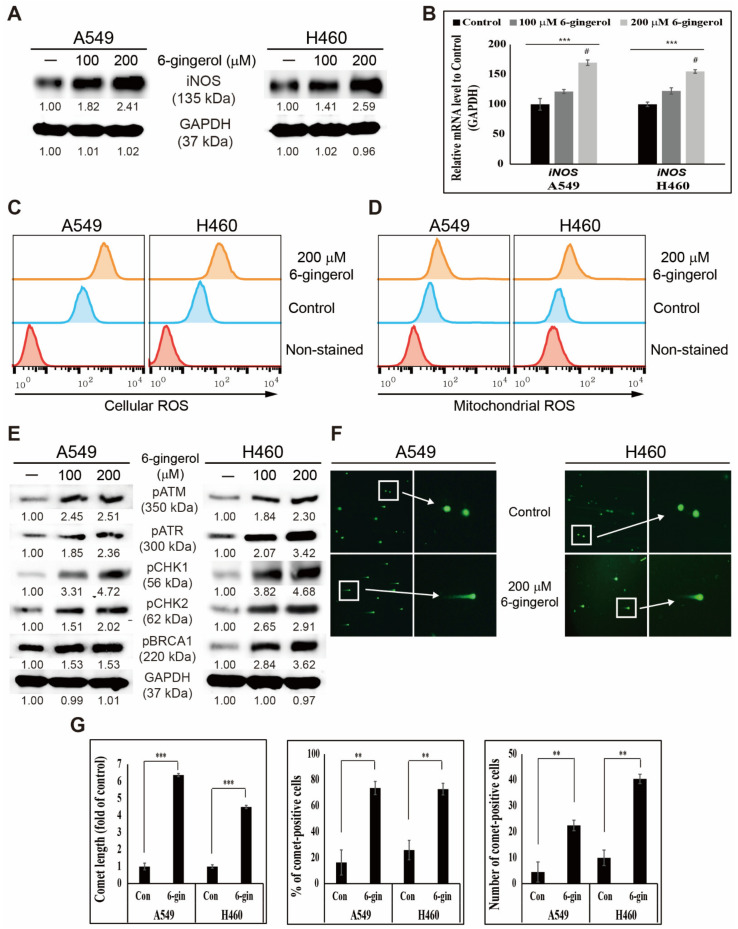
6-Gingerol induced ROS formation in A549 and H460 cells. (**A**) Western blotting of iNOS protein expression was incubated with 100 or 200 μM 6-gingerol for 48 h. (**B**) qRT-PCR showing illustrative iNOS expression. Next, controls were set to 100. *** *p* < 0.001 (ANOVA); # *p* < 0.001 vs. control. (**C**,**D**) Flow cytometry assay performed for detecting cellular and mitochondrial ROS. A plot of the percentage of cells with cellular and mitochondrial ROS induction. (**E**) The results of Western blotting indicate pH2AX, pATM, pATR, pCHK1, pCHK2, and pBRCA1 protein expression. (**F**) Comet assay images from fluorescent microscopy analysis at ×10 and ×40 magnification show fragmented DNA migration from the nucleoid body that forms a comet tail. (**G**) Graphical representation of the comet length analyzed as the fold change versus the control and percentage of comet-positive cells with 6-gingerol treatment in lung cancer cells. ** *p* < 0.01; *** *p* < 0.001 (Student’s *t*-test).

**Figure 2 cells-12-02628-f002:**
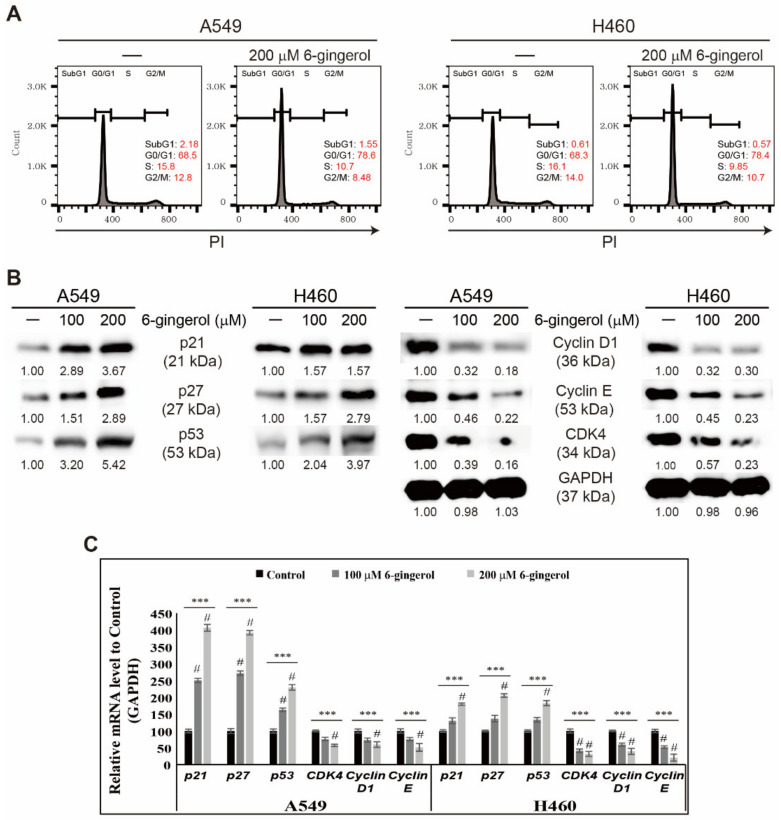
6-Gingerol induced cell cycle arrest in A549 and H460 cells. (**A**) Flow cytometry analysis using PI staining showed cell cycle distribution after 48 h treatment with 200 µM 6-gingerol. (**B**) Western blotting of A549 and H460 cells after 48 h treatment with 100 and 200 µM 6-gingerol showing p21, p27, p53, cyclin D1, cyclin E, and CDK4 expression. (**C**) qRT-PCR results show expression of cell cycle checkpoint genes. Representative cyclin D1, cyclin E, CDK4, p21, p27, and p53 mRNA expression is shown. Controls were set to 100. *** *p* < 0.001 (ANOVA test); # *p* < 0.001 vs. control.

**Figure 3 cells-12-02628-f003:**
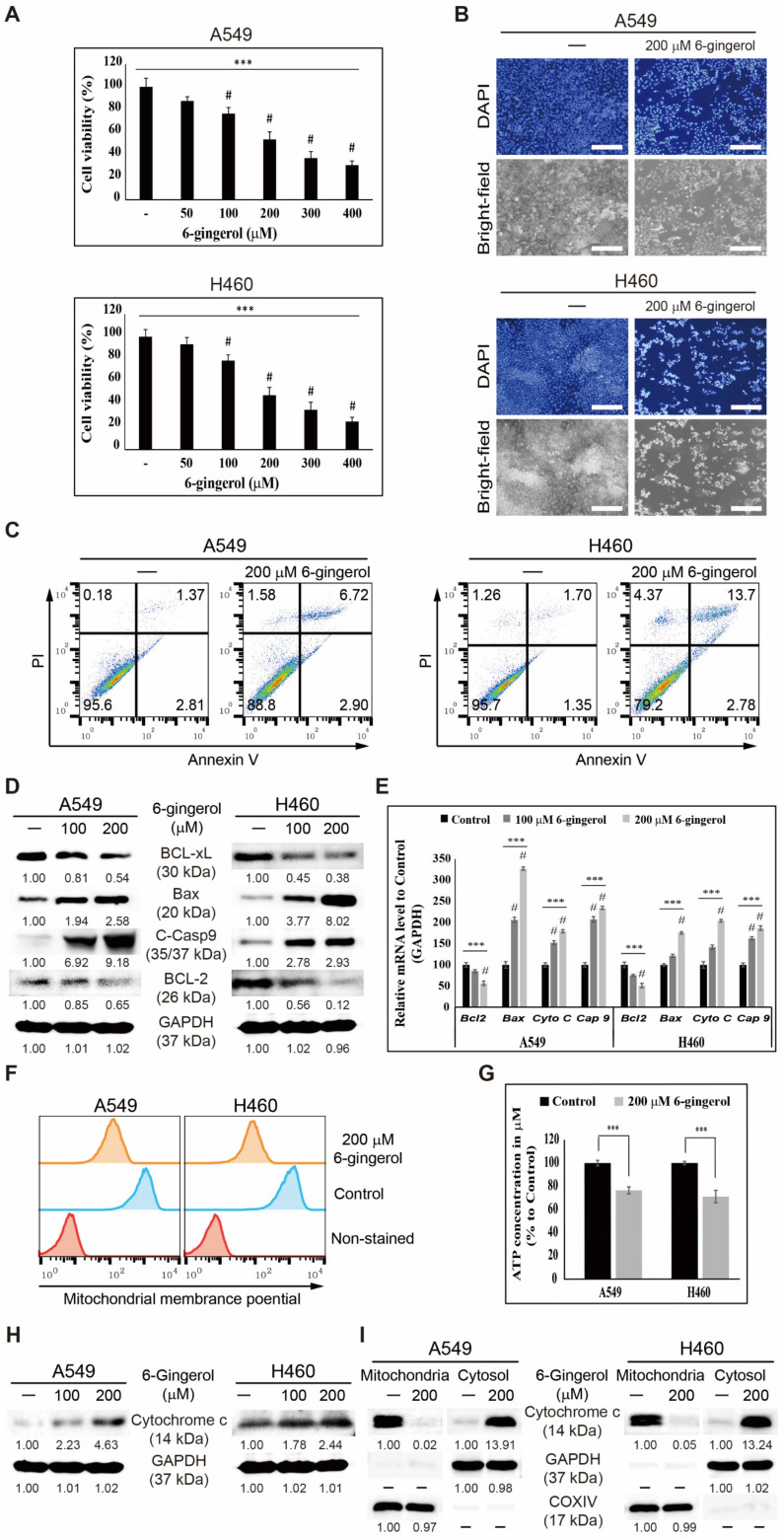
6-Gingerol induced the intrinsic apoptosis pathway in A549 and H460 cells. (**A**) Cell death analysis via the MTT assay in A549 and H460 cells after treatment with increasing concentrations of 6-gingerol for 48 h. *** *p* < 0.001 (ANOVA test); # *p* < 0.001 vs. control. (**B**) 6-Gingerol induced nuclear aberrations in NSCLC cells. DAPI staining and phase-contrast microscopy images show abnormal nucleus formation induced by 48 h treatment with 200 µM 6-gingerol. Representative photographs are presented. Scale bar 200 μm. (**C**) Annexin V-FITC vs. PI staining analysis after incubation with 200 μM 6-gingerol for 48 h. (**D**) Western blotting of BAX, BCL-2, BCL-xL, and C-Casp9 proteins. (**E**) qRT-PCR showing illustrative BAX, BCL-2, and caspase-9 gene expression. Next, Cp values were normalized to *GAPDH* mRNA. Controls were set at 100. *** *p* < 0.001 (ANOVA test); # *p* < 0.001 vs. control. (**F**) Flow cytometry analysis for mitochondrial membrane potential. (**G**) Plot of ATP concentration after treatment with 200 µM 6-gingerol. *** *p* < 0.001 (Student’s *t*-test). (**H**) Western blotting showing cytochrome c expression. (**I**) Western blotting of cytochrome c in cytosolic and mitochondrial fractions. GAPDH and COX IV were the controls for cytosolic and mitochondrial fractions, respectively.

**Figure 4 cells-12-02628-f004:**
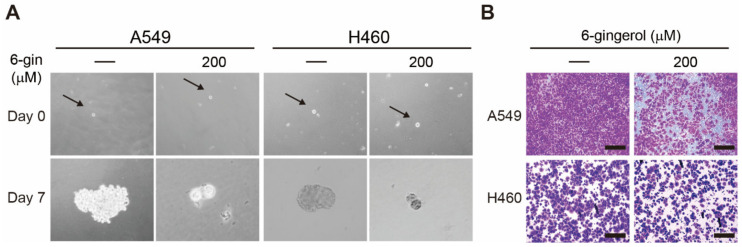
6-Gingerol inhibited tumor sphere formation and invasion. (**A**) A549 and H460 cells were cultured in DMEM/F-12 medium containing epidermal growth factor, basic fibroblast growth factor, and B27 for 7 days. Images were taken on days 0 and 7, and the tumor sphere increased in size in nontreated cells but decreased in those treated with 200 µM 6-gingerol. The arrow indicates a single cell. (**B**) Matrigel invasion assay illustrated that 200 µM 6-gingerol for 48 h could inhibit the invasion of A549 and H460 cells. Scale bar 100 μm.

**Figure 5 cells-12-02628-f005:**
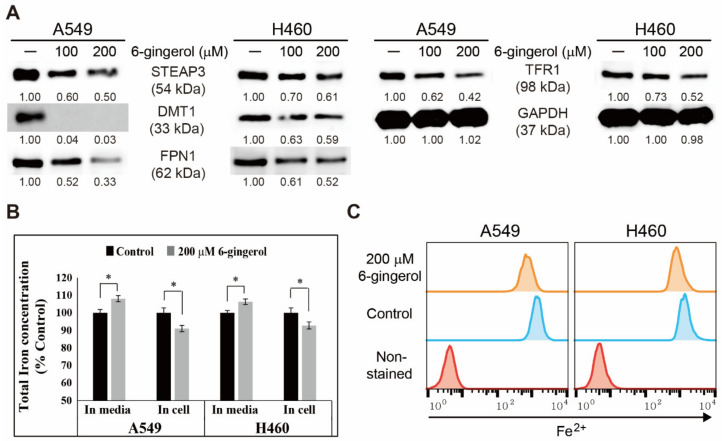
6-Gingerol inhibited iron metabolism in A549 and H460 cells. (**A**) Western blotting of TFR1, DMT1, STEAP3, and FPN1 after incubation with 100 or 200 μM 6-gingerol for 48 h. (**B**) Iron assay of the total iron concentration. * *p* < 0.1 (control vs. 6-gingerol; Student’s *t*-test). (**C**) Flow cytometry of Fe^2+^ in A549 and H460 cells after treatment with 200 μM 6-gingerol for 48 h.

**Figure 6 cells-12-02628-f006:**
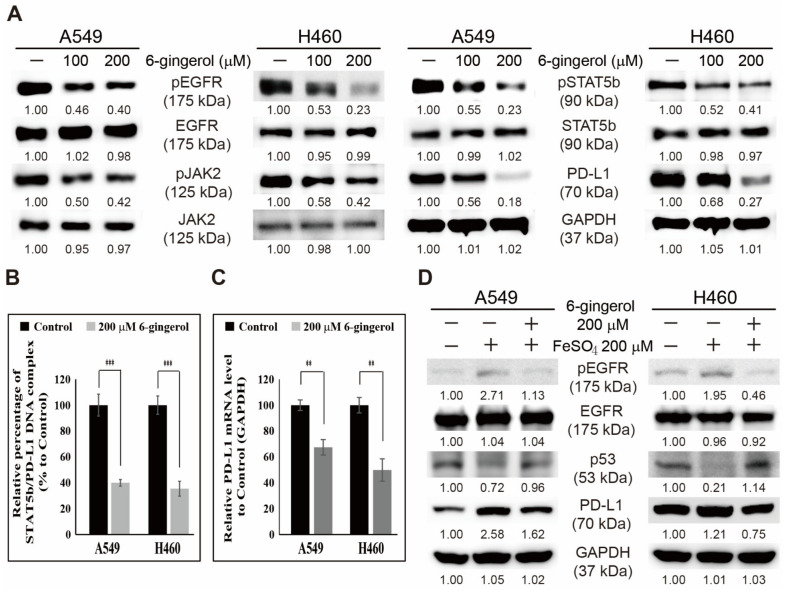
6-Gingerol inhibited EGFR/JAK2/STAT5b/PD-L1 signaling in A549 and H460 cells. (**A**) Western blotting of after 48 h treatment with 100 and 200 µM 6-gingerol showed pEGFR, EGFR, pJAK2, JAK2, pSTAT5, STAT5b, and PD-L1 protein expression. (**B**) ChIP assay analysis after A549 and H460 cells were incubated with 200 µM 6-gingerol for 48 h showed suppression of STAT5 binding to the PD-L1 promoter. The relative binding of STAT5 to the PD-L1 gene promoter was expressed as a percentage of control. Statistical analysis was performed using Student’s *t*-test (*** *p* < 0.001). (**C**) The expression levels of PD-L1 mRNA were detected after 6-gingerol treatment. Statistical analysis was performed using Student’s *t*-test (** *p* < 0.01). (**D**) Western blotting of pEGFR, EGFR, p53, and PD-L1 in A549 and H460 cells incubated with 200 µM FeSO_4_ or 200 μM 6-gingerol for 48 h.

**Figure 7 cells-12-02628-f007:**
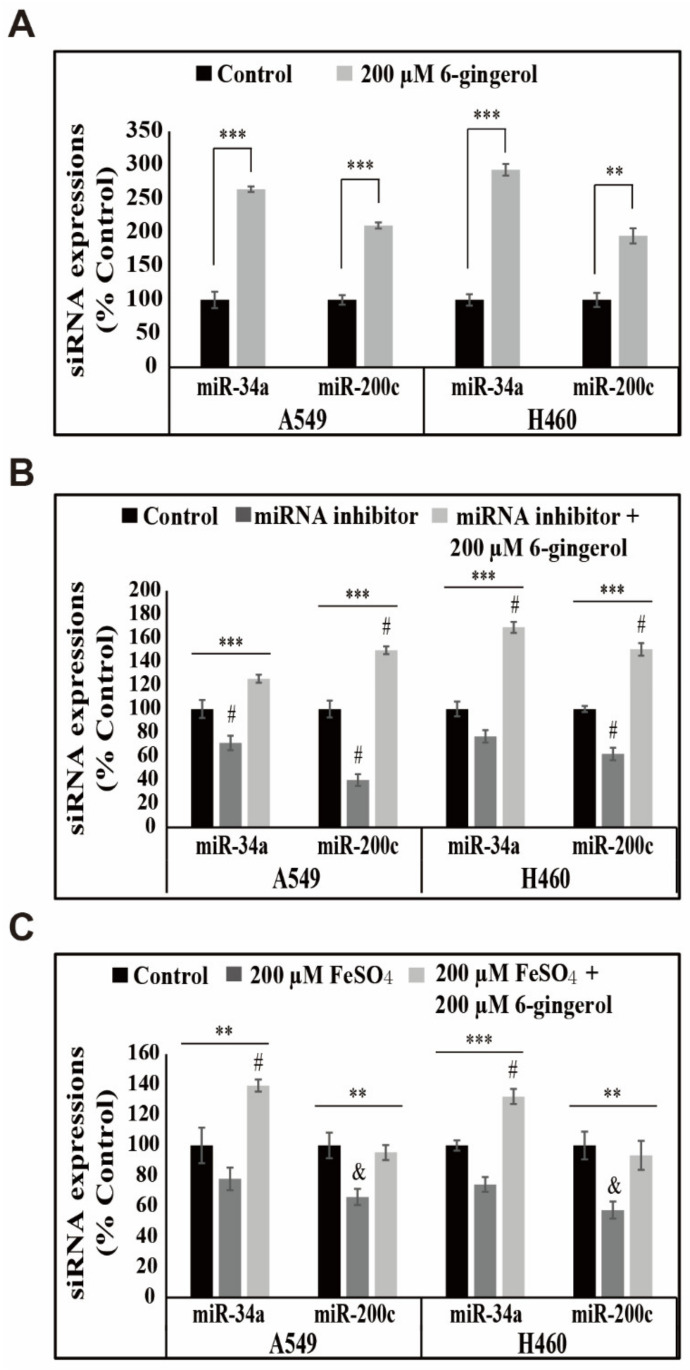
6-Gingerol regulated miR-34a and miR-200c expression in A549 and H460 cells. (**A**) Representative qRT-PCR of miR-34a and miR-200c transcripts. Cp values were normalized to U6 mRNA. Controls were set to 100. ** *p* < 0.01; *** *p* < 0.001 (Student’s *t*-test). (**B**) Representative qRT-PCR of miR-34a and miR-200c transcripts after treatment with an miRNA inhibitor with or without 200 μM 6-gingerol for 48 h. Cp values were normalized to *U6* mRNA. Controls were set to 100. *** *p* < 0.001 (ANOVA test); # *p* < 0.001 vs. control. (**C**) Representative qRT-PCR of miR-34a and miR-200c in A549 and H460 cells treated with 200 μM 6-gingerol for 48 h, followed by 200 µM FeSO_4_ for an additional 48 h. Cp values were normalized to *U6* mRNA. Controls were set to 100. ** *p* < 0.01; *** *p* < 0.001 (ANOVA test); ^&^
*p* < 0.01 vs. control; # *p* < 0.001 vs. control.

**Figure 8 cells-12-02628-f008:**
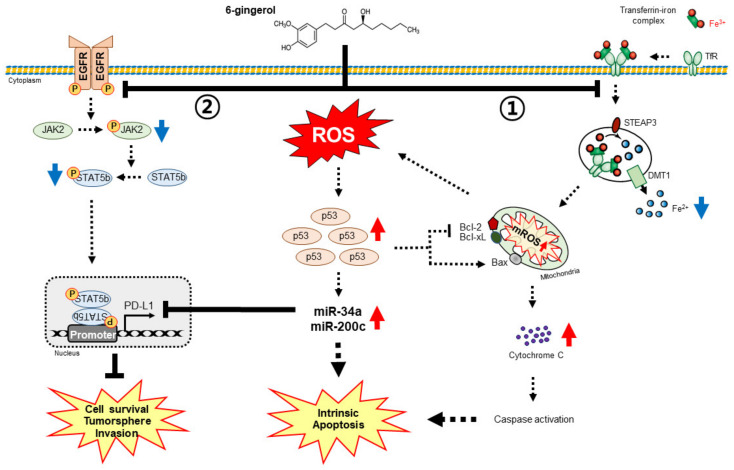
Molecular regulatory mechanism of mitochondrial apoptosis by 6-gingerol in NSCLC cells by regulating iron homeostasis (①) and EGFR/JAK2/STAT5b signaling pathway (②) inhibition, thereby suppressing PD-L1 expression through miR34a/miR-200c expression by 6-gingerol.

## Data Availability

The data presented in this study are available on request from the corresponding author. The data are not publicly available due to personal reasons.

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
