# Peer review of "Anticancer Effects of 6-Gingerol through Downregulating Iron Transport and PD-L1 Expression in Non-Small Cell Lung Cancer Cells"

_cells, 2023, doi:10.3390/cells12222628_

Round 1

Reviewer 1 Report

Comments and Suggestions for Authors

The manuscript by Kang et al. presents a characterization of the anticancer effects of 6-gingerol in two NSCLC cell lines. Most of the included experiments and the associated data analyses are carefully performed and appear sound. However, some places absolutely need correction and the addition of additional information is necessary to improve the quality of the manuscript.

Major concerns:

1.       The title is overinterpreting the actual data presented. The manuscript nicely shows the following points: that 6-gingerol possess anticancer activity, that 6-gingerol affects iron transport, that 6-gingerol affects miR34a/200c expression, and that 6-gingerol repress PD-L1 protein expression. However, it is not shown that the anticancer effects are a consequence of changed regulation in this cascade of events with PD-L1 being the major mediator of the anticancer effect. Before the given linking of observations in a cascade model with PD-L1 being the anticancer effector is solid many additional experiments are needed. Alternatively, the title should be changed to reflect what is shown with experimental data in the manuscript instead of the now-given overinterpretation. In this line, it should be specified that Figure 8 only is a model possibly integrating different observations, line 33 removed/modified as it is an overinterpretation (will require e.g. reporter assays and 3’UTR mutation analyses beforehand actually shown), lines 383-384 removed/modified as it is an overinterpretation, and lines 387-389 removed/modified as it is an overinterpretation.

2.       In Figure 6 it is shown that 6-gingerol represses PD-L1 protein expression. Since this is a key observation in the manuscript It could be very interesting to see the effect of 6-gingerol also on PD-L1 mRNA expression (a quantitative assay such as RT-qPCR). Many of the other proteins/genes examined in the manuscript were analyzed on both protein and mRNA levels. Thus, this experiment should be relatively simple given that the needed biological samples already exist. The result could be important concerning the postulated effect of the changed miR34a/200c expression to drive the altered PD-L1 protein expression.

3.       Figure 7B illustrates the effect of a used mRNA inhibitor. No details of the nature of this inhibitor are present in the materials and methods or result sections. This is required. Moreover, it could be optional to see the effect of this inhibitor on PD-L1 protein and mRNA expression to further substantiate that changes in miR34a/200c correlate with the changes in PD-L1 expression.

4.       The study is based on an analysis of the two NSCLC cell lines A549 and H460. These harbor KRAS mutations. This should be specified. Moreover, it should be specified in the results and discussion that the given observations are based on models with this genetic background and that a limitation of the study is the lack of analysis of cell lines with other genetic backgrounds commonly observed in NSCLC i.e. EGFR mutations, and thereby lack of translation of the results to all NSCLC cells.

Minor concerns:

1.       The abstract describes the regulatory effects of 6-gingerol. It could be more precise if some of these effects were further described in terms of the direction of the regulation (up or down!)  i.e. EGFR signaling, PD-L1 expression, etc.

2.       Line 198 Please specify the abbreviation mROS.

3.       Figure 1G Please indicate the number of cells analyzed for the presented data.

4.       Figure 2 The legend annotations are incorrect concerning the panels in the figure. Moreover, Figure Panel 2B is lacking according to the text of the legend.  

Comments on the Quality of English Language

No major comments to the English writing

Author Response

We really appreciate Reviewer #1’s feedback and constructive criticism for our revision. We have attached the Review Report as an attachment. We really appreciate it.

Reviewer 2 Report

Comments and Suggestions for Authors

The paper is well written, and presents results of a carefully designed series of experiments aimed at checking effects of 6-gingerol on NSCLC cells. However, several issues need to be dealt with before accepting it for publication:

11. The concentration of 6-Gingerol (200 μM) seems to be quite high. Before drawing any conclusions about its potential as a prospective drug candidate one should check its effects on normal cells. In other words – wouldn’t it kill all cells indiscriminately at this concentration?

22.  The Authors claim that in their experiment the concentration of 6-Gingerol (200 μM) induced ~50% cell death against both cell lines – however, I failed to find a figure supporting that – no figure shows respective quantified assay results (it seems that Fig. 3 would be the best place for it)

33.  Taking into account the pathways activated in the experiment, I wonder why the Authors do not consider ferroptosis, instead of apoptosis? On the other hand, in the experiments Annexin V Conjugates for Apoptosis 149 Detection kit was used to prove entering the apoptosis pathway – some more detailed discussion of that is needed

44.  The Authors should refer to other, similar works, in particular Thangavelu et al., Chemistry and physics of lipids, 245, 105206, (2022) – what is the novelty of their results compared to that paper?

55.   Surprisingly, the Authors do not relate to the previous work published by their group - (Kim et al., . Carcinogenesis, 35(3), 683–691 (2014)). Definitely, that paper focused on another compound in more detail but it seems that the research on 6-gingerol was initiated then, so it should be cited. Another work of interesr Rafieepour et al., Toxicology and industrial health, 35(11-12), 703–713 (2019)

Author Response

We really appreciate Reviewer #2’s feedback and constructive criticism for our revision. We have attached the Review Report as an attachment. We really appreciate it.

Round 2

Reviewer 1 Report

Comments and Suggestions for Authors

The authors have done an impressive work to fulfill the concerns from the reviewers. All concerns are now clearly addressed.

Reviewer 2 Report

Comments and Suggestions for Authors

I am satisfied by the Authors' responses and the updated manuscript.